# How Urban-Level Credit Expansion Affects the Quality of Green Innovation: Evidence from China

Zhengge Song, Jingjing Tang *, Haijian Zeng and Fangying Pang

School of Economics, Guangxi University, Nanning 530004, China; zhengge816@sina.com (Z.S.);
zenghj06@126.com (H.Z.); pfysylvia@hotmail.com (F.P.)
* Correspondence: tanggreen@sina.com

**Abstract:** We take the economic stimulus package in China as a quasi-natural experiment to investigate the effect of urban credit expansion on the quality of green innovation at the city level. The analysis takes urban-level and firm-level data from 2004 to 2015 and adopts the PSM-DID approach. Our empirical results suggest that the implementation of credit expansion makes a significant contribution to the improvement of green patent quality. In addition, the mechanism suggests that urban credit expansion policies promote corporate green innovation through channels such as providing credit expansion and a lower cost of financing enterprise transformation and upgrading. This research also suggest that credit expansion promotes economic growth while also incentivising first-tier cities to engage in more green transformations and upgrade to improve the quality of green patents. Our findings also provide an important insight for the implementation of credit expansion policies and the achievement of sustainable development in countries around the world, particularly in developing countries. Finally, this paper argues that China's credit expansion policy in 2009 has played a role in improving the quality of green innovation and improving green transformation.

**Keywords:** urban credit expansion; green patents quality; economic stimulus package; green transformation; PSM-DID method

## 1. Introduction

As a result of the global financial crisis caused by the subprime mortgage crisis in the United States, the global economy was plunged into a period of recession in 2008. Even the governments of developed countries had to come up with packages to bail out their economies and societies. The Chinese central government released a series of macroeconomic policies to alleviate the negative impacts of the financial crisis and credit expansion to revive market dynamics, with a "proactive" fiscal policy and a "moderately loose" monetary policy, that is, the "4 Trillion Stimulus Package" plan. Many scholars believe that the CNY 4 trillion investment programme achieved good results in stimulating the economy and restoring growth. For example, Zhou et al. [1] argued that the programme played a vital role in maintaining stable and rapid economic growth, with GDP growth reaching 9.6% and 9.1% in 2008 and 2009, respectively.

On the one hand, the Development Research Centre of the State Council of China and the World Wide Fund for Nature (WWF) jointly released the report "The Impact of China's Economic Stimulus Plan on Climate and Energy" in 2011, which revealed that 81% of the CNY 4 trillion credit expansion had been invested into new housing and infrastructure, leading to the growth of energy-consuming industries such as steel, non-ferrous metals, and cement; there is no doubt that this led to a heavy impact on the environment in the short term. Specially, according to the report, the CNY 600 billion investment into railways has led to the consumption of 28.3 million tonnes of iron and steel and 120 million tonnes of cement, which translates into an energy consumption of 30.6 million tonnes of standard coal. Meanwhile, the 600 billion CNY investment into highways has led to the consumption of 15 million tonnes of iron and steel, 133.5 million tonnes of cement, and 28.35 million

tonnes of asphalt, which translates into an energy consumption of 29.257 million tonnes of standard coal. Furthermore, Hao et al. [2] revealed that the pollutant emissions in China have been rising, and air quality has further deteriorated since 2010. On the other hand, as high investments in energy-intensive sectors are unsustainable, coupled with increasing investments in energy efficiency and emissions reduction, green innovation is expected to be promoted (14.5% of green investments). The report also indicated that each unit of central financial expenditure on energy saving could drive ten units of social investment (i.e., the driving factor is 10). This report also anticipated that, in the next few years after 2011, this would lead to a social investment of CNY 407.9 billion, and these gradual investments into the market would promote the long-term rapid development of the energy-saving industry. And the statistics on domestic lending by financial institutions by industry for 2009 are presented in Table 1.

**Table 1.** Statistics on domestic loans made by financial institutions by industry in 2009.

| Industry | Amount (Yuan Billion) | Proportion |
|---|---|---|
| Manufacturing industries | 62,387.55 | 25.02% |
| Water, environment and utilities management | 32,349.56 | 12.97% |
| Electricity, gas and water production and supply industry | 28,151.40 | 11.29% |
| Transport, storage and postal services | 27,783.55 | 11.14% |
| Real estate industry | 24,335.81 | 9.76% |
| Wholesale and retail trade | 22,871.59 | 9.17% |
| Leasing and business services | 18,341.09 | 7.35% |
| Construction industry | 9046.97 | 3.63% |
| Mining industry | 7739.67 | 3.10% |
| Agriculture, forestry, animal husbandry and fisheries | 5234.21 | 2.10% |
| Accommodation and catering | 2277.63 | 0.91% |
| Information transmission, computer services and software | 2176.39 | 0.87% |
| Residential and other services | 2164.27 | 0.87% |
| Public administration and social organisations | 1338.77 | 0.54% |
| Education industry | 981.54 | 0.39% |
| Financial industry | 622.63 | 0.25% |
| Culture, sports and recreation | 607.01 | 0.24% |
| Scientific research, technical services and geological survey industry | 503.05 | 0.20% |
| Health, social security and social welfare | 484.30 | 0.19% |
| Total | 249,397 | 100% |

Note: Data from Wind database.

Moreover, the report predicted that the impact of the stimulus package on China's energy efficiency and emissions reduction would turn from negative to positive after 2014, and the credit expansion would lead to about 270 million tonnes of emissions reduction capacity annually, reducing China's carbon dioxide emissions from the model's theoretical projection of 9.63 billion tonnes to 9.36 billion tonnes by 2020. Furthermore, many industries and regions in China still rely on the development style of high input, high emissions, and high pollution in pursuit of rapid economic growth, resulting in increasingly serious environmental concerns and unsustainable development [3]. In general, such environmental emergencies require rapid and revolutionary transformations in the way that human beings produce. Accordingly, incentivising firms to make a green transition while ensuring economic efficiency has become a pressing challenge for rapidly industrialising developing countries, with green innovation being considered a radical engine for promoting green productivity growth [4].

Generally, green innovation has been defined in [5] as innovation in technology, products, services, management models, or organisational structures to realise sustainable development, thus significantly minimising negative consequences for the environment. Recently, Hsu et al. [6] argued that green innovation combines scientific and technological innovation with environmental protection. Global environmental restoration is imminent [7], and green innovation helps address this dilemma, promoting both economic growth and environmental protection [8]. As environmental sustainability and economic profitability are equally important [9], green innovation leads to a sustainable competitive advantage and contributes to meeting the societal demand for the protection of the environment [10]. In [11,12], it has been suggested that green innovation is critical to enhance the sustainable development and environmental protection of enterprises (green competitiveness). Furthermore, the quality of the innovation leads to increased cost efficiency and organisational flexibility, which both contribute to the mitigation of environmental risks [13] and enhance efficiency in the utilisation of resources [14–16]. Accordingly, promoting green technological innovation and stimulating the vitality of green innovation in enterprises provides a pathway for green economic development. Nevertheless, green innovation is characterised by strong externalities, high investment, and high risk, and profit-maximising enterprises are often reluctant to engage in green innovation without external policy intervention. Therefore, it is necessary to investigate how to improve the sustainability of corporate green innovation.

The previous literature on credit expansion has focused on its negative impact on economic growth, firm performance, and household investment and financing [17–19]. Some studies have focused on the impact of the green credit policy on green innovation using firm-level data [20,21]. However, it remains unclear whether credit expansion has a long-term impact on green innovation at the firm level. In other words, whether credit expansion—as distinguished from green credit policies—can likewise stimulate green innovation in firms is still an open question. As such, the inspiration for this paper is that the motivations and purposes of these two types of credit expansion policies differ. Overall, this paper is the first to investigate how credit expansion affects the quality of green innovation among listed firms in the world's largest emerging market. Unlike previous studies, which have focused on green credit and green innovation [20,22,23], we consider another perspective and use city-level data to study how credit expansion affects the quality of green innovations in an attempt to find evidence that credit expansion is not only effective for expanding domestic demand and accelerating economic and industrial development but also has a positive impact on the quality of green innovation by enterprises.

This study attempts to demonstrate another perspective for exploring whether credit expansion policies in extraordinary times can similarly impact green innovation at the firm level. China's GDP growth has shown a linear growth trend, while China's invention patent application has shown an exponential growth trend since 2010, combined with the patent application situation [24]. With the support of credit funds, enterprises increase their investment in R&D in order to produce better-quality patents [25]. We use the difference-in-difference (DID) method and consider the "economic stimulus package" in 2009 as an exogenous change in policy to examine this relationship by utilising panel data from China within the period from 2004 to 2015. Finally, our findings suggest that the urban credit expansion had a significant positive effect on the quality of green patents, and the robustness test also confirms this result.

This paper contributes to the literature in three ways. First, to the best of our knowledge, this research is the first study to explore the relationship between urban credit expansion and the quality of green patents using city-level data, providing a new perspective to review the 2009 economic stimulus package. In other words, previous studies have mostly focused on the macro qualitative analysis of green financial instruments or green credit. In comparison, our study examines the effect of credit expansion policy on the quality of innovation from a city-level perspective. Second, we contribute to enriching the macroeconomic literature by examining the effect of credit expansion on green innovation

in the long term. Third, this paper reveals that urban credit expansion has a different effect on the quality of green patents in different tier cities and at different firm sizes, which also fills a gap in the current research on green finance. Finally, we provide policy recommendations based on our research findings, helping governments and financial institutions to better formulate and adjust credit policies to promote the positive impact of urban credit expansion on environmental protection and green transformation. Specifically, governments and regulators can increase their financial support for green innovation, particularly by encouraging more innovation through credit expansion. Furthermore, financial institutions and investors can develop green technology-related financial products, such as green bonds, to attract more investors to the sustainable development sector.

The remainder of this paper is structured as follows. In Section 2, we review the previous relevant research and propose the hypotheses of this study. In Section 3, we introduce the data sample, key variables, and research method. Section 4 presents the empirical findings, and Section 5 details the robustness test. Finally, Section 6 summarises the discussion and conclusion.

## 2. Literature Review and Hypothesis Development

### 2.1. Credit Expansion and Corporate Innovation

A well-developed and well-functioning financial system is critical for stimulating innovation and long-term economic growth [26]. Such mechanisms may typically arise through two mechanisms, with purely quantitative effects occurring when financial intermediaries convert savings into investments [27]. Additionally, the financial system can enhance the productivity of investment by allocating funds to the most qualified firms [28,29]. In [30], the impact of credit expansion on entrepreneurship was investigated. It was found that banks are less willing to bail out underperforming firms after deregulation and that firms in industries with a higher dependence on banks are more likely to engage in restructuring activities. Likewise, [31] also focused on credit expansion and entrepreneurship and found similar results. Subsequently, academics have begun to note the relationships between bank deregulation and firm innovation. In [32], it was shown that interstate banking deregulation increases the level of innovation and risk of young private firms, while intrastate deregulation reduces the level of innovation and risk. In addition, [33] provided new evidence that interstate banking deregulation has significantly affected the quantity and quality of innovative activity. In [34], it was suggested that the expansion of credit availability guides enterprises to improve their inputs and outputs for a given level of inputs. Other contemporaneous studies on credit availability and innovation have revealed mixed effects, depending on the type of deregulation [35,36]. In addition, Xin et al. [37] showed that long-term versus short-term bank lending and Chinese Big Four bank lending, compared to that of non-Big Four banks, have effects on technological innovation.

### 2.2. Green Finance and Green Innovation of Firms

The existing literature presents two different perspectives on the impact of green credit policies on firms' green innovation. Some studies have argued for the existence of a negative impact, as neoclassical economic theory suggests that environmental regulation increases the environmental expenditures of firms, thereby hindering their green innovation [38] as well as their willingness and ability to innovate [39,40]. In contrast, Porter's hypothesis [41] indicates that an adequate environmental regulation facilitates business innovation, as improvements in production technology resulting from innovation can partially or fully offset the costs of environmental protection. In addition, many scholars have also confirmed the validity of Porter's hypothesis through empirical evidence. Shive and Forster [42] found that the high cost of environmental regulation provides solid incentives for shareholders, as agents, to implement green-oriented corporate innovations. In [43], it was argued that green credit expansion in 2012 has significantly improves the quality of green innovation in polluting firms, which is well in line with the findings of [44,45]. However, Wang et al. [46,47] observed that while environmental regulation ham-

pers corporate innovation, the connection will become more accessible over time. Moreover, Chakraborty and Chatterjee [48] explored the indirect impact of environmental regulation on innovation activities in the Indian leather and textile industry and showed that green policies positively impact R&D investment and the patenting of technological innovations. Likewise, Yao et al. [49] confirmed that the scale of green credit in 30 provinces in China has a positive effect on the innovation of green technology.

The existing research suggests that green credit has a signalling effect due to its stimulating function [50], and the implementation of the green credit policy in China raises the cost of finance for heavily polluting firms [51] and decreases bank lending, capital investment, productivity, and operating performance in the short term [46,52]. The ability of the green credit policy to incentivise heavily polluting firms to develop green innovations through credit constraints is important, as it determines the validity of a company's green transition, particularly for heavily polluting companies with significant levels of energy consumption and pollution [53]. Moreover, Chen et al. [54] reported that green credit policies could significantly promote low-carbon technological innovation, with a more significant effect on low-carbon technological innovation in state-owned enterprises and ESG (environmental, social and governance)-certified firms. In [55], it was argued that firms with higher-quality environmental disclosures do not obtain more loans and that only green innovations can facilitate access to loans by firms. The root cause of this phenomenon is corporate "greenwashing," which is prevalent in soft environmental disclosures and prevents firms from obtaining more loans. Zheng et al. [56] concluded that the green credit policy has a significant facilitating effect on green innovation in heavy-polluting firms, and the supply of a local government environmental protection system can reinforce the effect of the green credit policy on green innovation; however, the institutional supply of innovations has not yet presented a significant positive effect.

### 2.3. Hypothesis Development

The report "The Impact of China's Economic Stimulus Plan on Climate and Energy" explains the effects of credit expansion on short-term and long-term energy conservation and emissions reduction outcomes. In particular, 81% of the funds were invested in large-scale infrastructure, stimulating the growth of high-energy-consuming industries, thus seriously impacting the ecological environment. Meanwhile, funds for energy conservation, reduction in emissions, and ecological construction reached CNY 210 billion, and other ecological and environmental investments have reached CNY 140 billion. Credit expansion provides more direct financial support, enabling enterprises to increase their R&D investment, thus contributing to increased green patent innovation quality. Moreover, credit expansion may make it easier for firms to access funds for R&D on green patents by lowering the cost of finance. Overall, many studies in the existing literature have suggested that credit expansion reduces the cost of finance for firms and makes it easier for firms to obtain finance [57]. Thus, easier credit conditions may make it easier for firms to access funds through traditional financing channels (e.g., bank loans). Therefore, this easier access to finance may allow firms to undertake larger and more complex green finance innovation projects, improving the quality of green innovation. Therefore, credit expansion may have such a mechanism of effect on the quality of corporate green patents.

**H$_1$:** *Urban credit expansion significantly promotes the quality of green patents.*

Due to the vastly different levels of infrastructure and economic development in different regions and cities, it is worth focusing on the heterogeneity of different cities. This study follows [58,59], classifying Chinese cities into five tiers. First-tier cities represent the most developed regions in China, with the highest level of economic development and abundant resources. These are densely populated metropolises, such as Beijing, Shanghai, Guangzhou, and Shenzhen, with enormous economic, cultural, and political influence in China. We grouped descriptive statistics for the variable Bl09 (the level of credit expansion in 2009) and observed that the average value of urban credit over-expansion in 2009 varied

across city tiers (as shown in Table 2). We find that the average number of citations to green patents within 5 years was the highest in first-tier cities (3.466). Moreover, previous studies have suggested that bank credit varies across cities [60,61]. In [62], evidence that green finance can significantly contribute to the transition to a low-carbon economy in China was provided, where this positive effect varies in different regions (e.g., the impacts in the West are not significant). On the other hand, considering the differences in innovation ecosystems, large cities typically have more developed innovation ecosystems [63], including research institutions, higher education institutions, and innovative firms. Thus, city-level credit expansion may be more conducive to building and strengthening green technology innovation ecosystems in larger cities, which may have more pronounced impacts on the quality of green patents. This study examines the effect of heterogeneity across different tiers of cities, and we argue that different levels of urban credit expansion also affect the quality of green patents in different tiers of cities, especially in first-tier cities (e.g., Beijing and Shanghai), where the average value of credit expansion is higher (see Table 2).

**Table 2.** Average urban credit expansion in 2009.

| Tier of Cities | Excessive Credit Expansion (bl09) | Cit5greenpant |
|:---:|:---:|:---:|
| Tier 1 | 0.263 | 3.466 |
| Tier 2 | 0.157 | 3.190 |
| Tier 3 | 0.137 | 3.100 |
| Tier 4 | 0.100 | 3.162 |
| Tier 5 | 0.108 | 3.223 |

**H$_2$:** *Urban credit expansion in first-tier cities has a significant positive impact on the quality of green patents compared to low-tier cities.*

Due to large differences in the internal controls and financing capacity between firms of different sizes [64], varying effects of credit expansion policies may be observed. As most large firms are state-owned in China [65], they need to respond to the policy guidelines of the Communist Party, and the 18th Party Congress in 2012 proposed to promote green innovation and the upgrading of traditional industries and vigorously develop a green economy. Therefore, large-sized firms have richer resources as well as policy guidelines to incentivise their green transformation and are more capable of and motivated to make green innovation changes. Large-sized firms may gain greater performance growth benefits from technological upgrading and have more incentive to pursue green innovation. In contrast, small-sized firms are more engaged in entrepreneurial production activities through credit expansion. In addition, larger firms are more likely to engage in corporate social responsibility [66] and therefore have a greater incentive to carry out green innovation during city-level credit expansion. Meanwhile, small and medium-sized enterprises may differ in their sense of social responsibility, and some may have difficulty integrating green innovation into their strategic planning due to limited resources. Hence, we argue that the impact of credit expansion on the quality of green patents varies at the firm level across different firm sizes.

**H$_3$:** *Urban credit expansion has a significant positive effect on green patent quality in large-sized firms compared to small-sized firms.*

## 3. Research Design

### 3.1. Data and Sample Selection

This paper focuses on the impact of credit expansion at the city-level in 2009 on the quality of green patents in Chinese listed companies. Thus, the core explanatory variable is the above-normal size of new credit at the city level, while the explanatory variables in this study are the quality of green financial patents, which were obtained from the CN Deep

database of firm-level data from 2004 to 2015. Specifically, green patents refer to patents whose IPC classification number belongs to the "Green Patent IPC Classification Number List." In general, the "Green Patent IPC Classification Number List" of patent classification numbers was developed by the Expert Committee on International Patent Classification Numbers (ECIPC), based on environmentally sound technologies (ESTs) listed in the United Nations Framework Convention on Climate Change (UNFCCC). At present, the green patents of Chinese listed companies are widely recognised as the main form of green innovation, and their information measures a company's green innovation capacity and output. The reason we chose this time span is that the year in which the exogenous policy (stimulus package) occurred was 2009, and too long a time span could introduce other uncontrollable variations and confounders that may reduce the interpretability and credibility of the findings. Therefore, we select dataset from 2004 to 2015 as the research sample. In addition, we also select the firms' characteristic variables which were sourced from the China Stock Market and Accounting Research database (CSMAR).

### *3.2. Variable Definition*

#### 3.2.1. Core Explanatory Variables

The core explanatory variable used in this paper is the level of credit expansion *AbnBL* in 2009. We follow the method of bank credit expansion published in [67], which uses the level of local bank credit in 2009 minus the average level over the past five years to obtain the size of the new local bank credit. This part of the unusually high level of new credit is used as a proportion of a city's GDP in 2009 in order to measure the extent of the city's credit expansion, which is calculated using the following equation:

$$AbnBL_{i,2009} = \frac{BankLoan_{i,2009}}{GDP_{i,2009}} - \frac{1}{5} \times \sum_{t=2004}^{2008} \frac{BankLoan_{i,t}}{GDP_{i,t}} \tag{1}$$

#### 3.2.2. Dependent Variables

In [68,69], it was stated that a number of green patents can be used to measure green innovation and test the impact of environmental regulations on green innovation. Nevertheless, in 2007, [70] argued that the importance of different patents varies greatly, and thus the measurement of green innovation according to the number of patents will be biased. Subsequently, Wang et al. [43] argued that green patent citations are more accurate in expressing a firm's level of green innovation than the quantity of green patents. Therefore, in order to explore the impact of urban credit expansion on the Chinese listed firms in terms of green patent quality, this study followed [43] and selected the average number of citations to green patents within a 5-year period of public disclosure (Cit5greenpant) as a proxy for green patent quality. Likewise, the average number of citations to green patents within a 7-year period (Cit7greenpant) was used as an additional dependent variable to examine the relationship between credit expansion and the quality of green patents. We set the dependent variable forward by one period to avoid the reverse causality problem.

#### 3.2.3. Control Variables

In terms of control variable selection, we followed [71] and selected a series of characteristic variables at the city level. These control variables included lngdp for the city's GDP per capita, measured using the natural logarithm of a city's GDP. Industry denotes the total share of secondary and tertiary industries in GDP. FAI represents the share of total local fixed asset investment in GDP. Investment denotes the share of total foreign investment in GDP, and lnpop is the population size at the end of the year, measured as the natural logarithm of the population. In addition, the volatility of a firm's green innovation output (volatility green) was measured using the standard deviation of the average number of citations within a 5-year period. Subsequently, we also added firm-level control variables, including firm size (size), which is the natural logarithm of a firm's total assets; firm profitability (profitability), calculated as the firm's net return divided by the book

assets; and book leverage (leverage), measured as the ratio of total liabilities to total assets. Furthermore, a firm's tangibility was proxied using their proportion of fixed assets to total assets. These firm-level variable data were sourced from the CSMAR database. For this study, non-missing values were required, with all variables winsorised at the 1st and 99th percentiles to remove the possible impact of extreme values. Overall, the final data sample consisted of 2158 observations within the period from 2004 to 2015 for empirical analysis.

*3.3. Methodology and Endogeneity Concern*

In the baseline regression process, since the division of the treatment and control groups is not randomly selected, and there are different characteristics of the treatment and control groups, this would cause a self-selection bias in the DID method, and such a bias would cause a correlation between the explanatory variables and the residual term, which would lead to an endogeneity problem. Specifically, due to the existence of heterogeneity in variables such as credit size and industry size between different cities, the empirical results may be biased and fail to reflect the real policy effects. The propensity score matching method (PSM) developed by [72,73] was adopted to address this concern and eliminate sample selection bias. In contrast to the DID method, DID-PSM takes into account individual differences in the experimental and control groups. Furthermore, the PSM can address the sample selection bias rather than remove endogeneity concerns caused by the omitted variables. Hence, the difference-in-difference model alleviates the endogeneity issue and derives policy treatment effects well but does not function well to solve the sample bias. Accordingly, we applied the PSM-DID estimation method, through which we first find the control group that is closest to the treatment group in terms of the control variables and then conduct a DID regression using the matched treatment and control groups. Therefore, we explored the relationships between regional credit expansion and the quality of green patents using the PSM-DID method to mitigate endogeneity and selection bias concerns. The regression model used in this study is structured as follows:

$$Y_{it+1} = \alpha + \beta_1 Treat_{it} \times Time_{it} + \lambda X_{it} + \mu_i + \gamma_t + \varepsilon_{it} \tag{2}$$

where $Y_{it}$ denotes the green patent quality, and the interaction term $Treat_{it} \times Time_{it}$ is between the treatment dummy variable and the policy year dummy variable. Specifically, this paper takes the median of the new excess credit as a benchmark: if a city above the median level is set as the treatment group, the *Treat* variable takes a value of 1, and if a city below the median level is set as the control group, it takes a value of 0. We took 2009 as the year of an exogenous change in policy, so the *Time* variable takes a value of 1 if the year is 2009 or after 2009 and 0 otherwise. $X_{it}$ denotes the control variables, with respect to which we introduce firm- and city-level indicators in this study. $\mu_i$ and $\gamma_t$ represent the time fixed effects and individual fixed effects, respectively, and $\varepsilon_{it}$ is an error term.

To further understand the share of the stimulus package received by different city tiers, we calculated the descriptive statistics according to different city tiers in our sample. From Table 2, it is clear that the mean value of credit expansion in 2009 was the highest among Tier 1 cities (0.263), indicating that Tier 1 cities received the largest share of the stimulus package compared to other (low-tier) cities. The average number of citations to green patents within 5 years was also the highest (3.466). Therefore, we tested the effect of urban credit expansion on green patent quality across urban tiers (see Section 4).

Tables 3 and 4 present the descriptive statistics and correlation matrix of the main variables, respectively. Specifically, Table 3 provides the mean and median values for the full sample of 2158 observations. The average number of citations to green patents within 5 years of publication was 3.338, with a standard deviation of 2.449, while the average number of citations to green patents within 7 years of publication was 4.032, with a standard deviation of 2.897, suggesting substantial differences among the firms. These results indicate that the mean value of the average number of citations within a 7-year period was higher than that within a 5-year period, and the standard deviations indicate that the average number of citations within 5 years was less volatile. The mean value

of urban-level credit expansion (Abnbl09) was 0.123, with a standard deviation of 0.079, indicating the lower volatility of urban credit expansion. Finally, Table 3 also provides the descriptive statistics of the remaining variables, while Table 4 shows no clear evidence that these variables were auto-correlated.

**Table 3.** Descriptive statistics.

| Variable | Obs | Mean | Std. Dev. | Min | Max |
|---|---|---|---|---|---|
| *Cit5greenpant* | 2158 | 3.338 | 2.449 | 0.800 | 14.500 |
| *Cit7greenpant* | 2158 | 4.032 | 2.897 | 1.000 | 16.000 |
| *Abnbl09* | 2158 | 0.123 | 0.079 | −0.006 | 0.411 |
| *lnpgdp* | 2158 | 9.988 | 0.755 | 8.420 | 11.719 |
| *Growth* | 2158 | 13.103 | 3.362 | 1.500 | 22.800 |
| *Fai* | 2158 | 0.604 | 0.236 | 0.214 | 1.469 |
| *Industry* | 2158 | 0.855 | 0.087 | 0.634 | 0.990 |
| *Investment* | 2158 | 0.003 | 0.004 | 0.000 | 0.029 |
| *lnpop* | 2158 | 5.840 | 0.651 | 3.970 | 7.005 |
| *Volatility green* | 2158 | 1.422 | 1.873 | −0.896 | 6.561 |
| *Tangibility* | 2158 | 0.241 | 0.164 | 0.009 | 0.622 |
| *Leverage* | 2158 | 0.435 | 0.195 | 0.052 | 0.925 |
| *Profitability* | 2158 | 0.058 | 0.047 | −0.123 | 0.202 |
| *Size* | 2158 | 22.497 | 1.615 | 19.907 | 27.547 |

**Table 4.** Correlation matrix.

| Variables | (1) | (2) | (3) | (4) | (5) | (6) | (7) | (8) | (9) | (10) | (11) | (12) | (13) |
|---|---|---|---|---|---|---|---|---|---|---|---|---|---|
| *Cit5greenpant* | 1.000 | | | | | | | | | | | | |
| *Abnbl09* | 0.036 | 1.000 | | | | | | | | | | | |
| *lnpgdp* | 0.089 | 0.154 | 1.000 | | | | | | | | | | |
| *Industry* | 0.028 | 0.203 | 0.741 | 1.000 | | | | | | | | | |
| *Fai* | 0.057 | −0.004 | −0.230 | −0.186 | 1.000 | | | | | | | | |
| *Growth* | 0.359 | 0.045 | −0.246 | −0.106 | −0.001 | 1.000 | | | | | | | |
| *Investment* | 0.048 | 0.164 | 0.173 | 0.171 | 0.096 | 0.019 | 1.000 | | | | | | |
| *lnpop* | 0.012 | 0.009 | −0.145 | −0.02 | 0.417 | −0.150 | 0.048 | 1.000 | | | | | |
| *Volatility* | 0.118 | −0.016 | −0.007 | 0.011 | 0.041 | 0.005 | 0.010 | −0.034 | 1.000 | | | | |
| *Tangibility* | −0.047 | 0.033 | 0.037 | −0.007 | −0.032 | −0.001 | −0.081 | 0.036 | −0.053 | 1.000 | | | |
| *Leverage* | 0.053 | 0.012 | 0.049 | 0.019 | −0.002 | 0.052 | −0.028 | −0.026 | −0.057 | 0.143 | 1.000 | | |
| *Profitability* | −0.038 | 0.023 | −0.059 | −0.004 | 0.032 | −0.004 | 0.109 | 0.041 | 0.028 | −0.030 | 0.136 | 1.000 | |
| *Size* | 0.028 | 0.036 | 0.062 | 0.032 | 0.018 | 0.067 | −0.086 | −0.034 | −0.007 | 0.292 | 0.441 | 0.180 | 1.000 |

## 4. Empirical Results

In the baseline regression, we first investigated the preliminary relationship between credit expansion and the quality of green patents through the fixed effect model to support the hypotheses of this study. The following equation was used:

$$Y_{t+1} = \alpha + \beta_1\, Bl09_t + \lambda X_t + Year_t + Firm_i + \varepsilon_{it} \tag{3}$$

where the dependent variable $Y_{t+1}$ on the left-hand side is the average number of citations to green patents within a 5- or 7-year period of public disclosure, the independent variable is the excess scale of urban bank credit accounting for its proportion in the urban GDP in 2009, $X$ represents the control variables in the model, $\alpha$ is a constant, $Year_t$ and $Firm_i$ are time and firm fixed effects, respectively, and $\varepsilon$ is the error term. The standard errors were robust and clustered at the city and province levels. The empirical results are detailed in the following.

Table 5 shows the fixed effect regression results of credit expansion on the quality of the green patents of listed firms. The baseline empirical results indicate that the coefficient of urban bank credit expansion with respect to the quality of green patents (average number of citations within a 5-year period) was 1.973, which was positive and significantly associated at the 1% statistical level. Hence, firms with greater access to city-level credit expansion resources had a higher green patent quality than firms with less access to credit in the 2009 stimulus package. Similarly, the coefficient of credit expansion with respect to the average number of citations to green patents within a 7-year period was 2.107, which was also positive and significant at the 1% level. These baseline regression results suggest that

the expansion of urban bank credit in 2009 positively affected the quality of green patents while promoting economic development.

**Table 5.** The effect of credit expansion on the quality of green patents.

| Dependent Variable | Cit5greenpant | | Cit7greenpant | |
|---|---|---|---|---|
| | **(1)** | **(2)** | **(3)** | **(4)** |
| *Bl09* | 1.043 *** | 1.973 *** | 1.061 * | 2.107 *** |
| | (0.514) | (0.680) | (0.604) | (0.748) |
| *lnpgdp* | | 0.030 | | −0.111 |
| | | (0.187) | | (0.212) |
| *Fai* | | 0.369 | | 0.156 |
| | | (0.325) | | (0.350) |
| *Growth* | | 0.032 | | −0.044 |
| | | (0.027) | | (0.030) |
| *Industry* | | −0.997 | | −0.064 |
| | | (1.291) | | (1.497) |
| *Investment* | | −0.767 | | 1.397 |
| | | (17.437) | | (16.565) |
| *lnpop* | | 0.049 | | 0.043 |
| | | (0.081) | | (0.094) |
| *Volatility* | | −0.127 ** | | −0.111 * |
| | | (0.057) | | (0.061) |
| *Tangibility* | | −0.346 | | −0.540 |
| | | (0.470) | | (0.543) |
| *Leverage* | | −0.420 | | −0.937 *** |
| | | (0.444) | | (0.528) |
| *Profitability* | | 0.304 | | −1.445 |
| | | (1.568) | | (1.703) |
| *Size* | | 0.175 *** | | 0223 ** |
| | | (0.084) | | (0.085) |
| Firm FE | YES | YES | YES | YES |
| Year FE | YES | YES | YES | YES |
| R-Squared | 0.088 | 0.125 | 0.132 | 0.103 |
| Observations | 2158 | 2158 | 2158 | 2158 |

Note: The standard errors in parenthesis and clustered at the city and province level, and significance level are *** $p < 0.01$, ** $p < 0.05$, * $p < 0.10$.

Table 6 presents the DID results regarding how credit expansion affected the quality of green patents. The coefficient of the DID interaction term was 0.466, with a significantly positive correlation at the 1% level, indicating that the average number of citations to green patents within 5 years was significantly higher in cities with more credit than in cities with lower credit after the implementation of the economic stimulus package. Likewise, the coefficient of the DID interaction term was 0.512, which was also a significantly positive correlation at the 1% level. These empirical results verify $H_1$ and prove that urban credit expansion promotes the quality of green innovation. Furthermore, cities that benefit from credit expansion programmes place greater emphasis on environmental protection and green innovation than other cities, promoting green transformation and the sustainable development of enterprises. Therefore, the 2009 stimulus package significantly increased the quality of green innovation, as the number of citations increased year by year, as shown by the coefficient of the average number of citations to green patents within 7 years.

**Table 6.** Difference-in-difference results.

| Dependent Variable | Cit5greenpant | | Cit7greenpant | |
|---|---|---|---|---|
| | **(1)** | **(2)** | **(3)** | **(4)** |
| *DID* | 0.378 ** | 0.466 *** | 0.399 ** | 0.512 *** |
| | (0.164) | (0.161) | (0.180) | (0.172) |
| *lnpgdp* | | −0.190 | | −0.232 |
| | | (0.202) | | (0.207) |
| *Fai* | | 0.162 | | 0.120 |
| | | (0.397) | | (0.419) |
| *Growth* | | 0.012 | | 0.010 |
| | | (0.031) | | (0.035) |
| *Industry* | | −0.890 | | −0.777 |
| | | (1.587) | | (1.740) |
| *Investment* | | 21.400 | | 28.215 |
| | | (21.988) | | (24.186) |
| *lnpop* | | −1.067 | | −0.166 |
| | | (0.134) | | (0.156) |
| *Volatility* | | −0.166 *** | | −0.183 *** |
| | | (0.057) | | (0.061) |
| *Tangibility* | | −1.031 * | | −1.161 * |
| | | (0.561) | | (0.632) |
| *Leverage* | | −0.064 | | −0.375 |
| | | (0.608) | | (0.641) |
| *Profitability* | | 0.872 | | −0.400 |
| | | (2.108) | | (2.109) |
| *Size* | | 0.151 | | 0.201 ** |
| | | (0.094) | | (0.097) |
| Firm FE | YES | YES | YES | YES |
| Year FE | YES | YES | YES | YES |
| R-Squared | 0.288 | 0.242 | 0.305 | 0.250 |
| Observations | 1968 | 1968 | 1968 | 1968 |

Note: The standard errors in parenthesis and clustered at the city and province level, and significance level are *** $p < 0.01$, ** $p < 0.05$, * $p < 0.10$.

*Heterogeneity Test*

Tables 7 and 8 report the results of the effect of urban credit expansion on the quality of green patents, according to the different tiers of cities. The coefficients for the average number of citations to green patents within 5 and 7 years were 2.376 and 1.928 for first-tier cities, which were significantly positive at the 1% and 10% levels, respectively. Meanwhile, the effect on low-tier cities was not significant, as shown in Tables 5 and 6, indicating that urban credit expansion had no significant effect on the quality of green patents in lower-tier cities. Possible reasons for this may include the fact that lower-tier cities are not as resource-rich as first-tier cities and are not as committed to the green transition, while other lower-tier cities were still at the stage of economic development, especially during the latter part of the economic crisis. Additionally, first-tier cities may be more efficient in resource allocation and can better utilize resources for city-level credit expansion. In contrast, small low-tier cities may face limited resources and need more targeted policy support to ensure that resources are used efficiently. Finally, these findings are similar to those reported in [8] and verify $H_2$ that urban credit expansion in first-tier cities has a significant positive impact on the quality of green patents compared to low-tier cities.

**Table 7.** The effect of credit expansion on the quality of green patents (city tier).

| Dependent Variable | Cit5greenpant | | | | |
|---|---|---|---|---|---|
| | Tier 1 | Tier 2 | Tier 3 | Tier 4 | Tier 5 |
| | (1) | (2) | (3) | (4) | (5) |
| DID | 2.376 *** | −0.293 | 0.540 | −0.201 | 0.403 |
| | (0.672) | (0.501) | (0.352) | (0.272) | (0.268) |
| lnpgdp | 0.632 | 0.461 | 0.180 | 0.276 | 0.090 |
| | (0.853) | (0.531) | (0.423) | (0.384) | (0.285) |
| Fai | −2.885 | −0.198 | 0.206 | −0.342 | −0.257 |
| | (4.360) | (1.569) | (0.898) | (0.615) | (0.497) |
| Growth | 0.373 *** | 0.021 | 0.091 | 0.012 | 0.029 |
| | (0.142) | (0.075) | (0.055) | (0.048) | (0.033) |
| Industry | −15.336 | 1.705 | 0.549 | −0.192 | 0.435 |
| | (13.632) | (6.958) | (3.510) | (3.136) | (2.142) |
| Investment | −12.753 | −9.638 | 41.719 | 45.646 *** | 2.793 |
| | (8.405) | (36.766) | (35.432) | (20.927) | (19.955) |
| Volatility | 0.363 | 0.151 *** | 0.031 | 0.102 | 0.127 * |
| | (0.350) | (0.083) | (0.115) | (0.119) | (0.066) |
| Tangibility | 2.771 | −0.837 | −0.450 | −0.408 | −0.397 |
| | (1.560) | (0.877) | (1.320) | (1.164) | (0.748) |
| Leverage | 4.547 *** | 3.076 ** | 0.757 | 1.525 * | 1.845 ** |
| | (1.877) | (1.242) | (0.890) | (0.805) | (0.764) |
| Profitability | 11.569 | 7.831 | 2.015 | 8.358 ** | 1.676 |
| | (10.843) | (5.385) | (2.624) | (3.734) | (3.170) |
| Size | −0.699 ** | −0.326 *** | −0.013 | −1.034 | −0.139 |
| | (0.299) | (0.085) | (0.178) | (0.135) | (0.113) |
| Firm FE | YES | YES | YES | YES | YES |
| Year FE | YES | YES | YES | YES | YES |
| R-Squared | 0.247 | 0.134 | 0.047 | 0.082 | 0.046 |
| Observations | 126 | 274 | 479 | 452 | 827 |

Note: The standard errors in parenthesis and clustered at the city and province level, and significance level are *** $p < 0.01$, ** $p < 0.05$, * $p < 0.10$.

**Table 8.** The effect of credit expansion on the quality of green patents (city tier).

| Dependent Variable | Cit7greenpant | | | | |
|---|---|---|---|---|---|
| | Tier 1 | Tier 2 | Tier 3 | Tier 4 | Tier 5 |
| | (1) | (2) | (3) | (4) | (5) |
| DID | 1.928 * | −0.599 | 0.461 | −0.419 | 0.209 |
| | (0.860) | (0.548) | (0.455) | (0.317) | (0.308) |
| lnpgdp | 0.697 | 0.221 | −0.349 | −0.268 | −0.343 |
| | (0.764) | (0.651) | (0.535) | (0.448) | (0.303) |
| Fai | −3.505 | −1.037 | −0.358 | −1.169 | −0.934 |
| | (5.088) | (1.863) | (1.059) | (0.753) | (0.556) |
| Growth | 0.437 ** | 0.108 * | 0.126 ** | 0.032 | 0.058 |
| | (0.152) | (0.060) | (0.060) | (0.047) | (0.045) |
| Industry | −12.810 | 8.251 | 2.539 | 1.118 | 2.264 |
| | (18.251) | (8.782) | (3.643) | (4.062) | (2.243) |
| Investment | −14.040 | −7.838 | 54.415 | 58.759 ** | 12.472 |
| | (9.293) | (45.350) | (41.552) | (26.561) | (20.388) |
| Volatility | 0.523 | 0.298 ** | 0.150 | 0.264 | 0.205 ** |
| | (0.345) | (0.105) | (0.141) | (0.153) | (0.082) |
| Tangibility | 3.347 | −1.221 | −0.118 | −0.223 | 0.403 |
| | (1.940) | (1.035) | (1.483) | (1.281) | (0.820) |
| Leverage | 4.209 | 2.985 ** | 0.590 | 1.271 | 1.864 ** |
| | (2.402) | (1.354) | (0.973) | (1.014) | (0.842) |

**Table 8.** *Cont.*

| Dependent Variable | Cit7greenpant | | | | |
|---|---|---|---|---|---|
| | **Tier 1** | **Tier 2** | **Tier 3** | **Tier 4** | **Tier 5** |
| | **(1)** | **(2)** | **(3)** | **(4)** | **(5)** |
| *Profitability* | 12.825 | 7.370 | 2.610 | 7.283 | 2.254 |
| | (11.337) | (5.828) | (3.000) | (7.411) | (3.268) |
| *Size* | −0.740 ** | −0.321 *** | 0.062 | −0.093 | −0.010 |
| | (0.319) | (0.105) | (0.202) | (0.118) | (0.146) |
| Firm FE | YES | YES | YES | YES | YES |
| Year FE | YES | YES | YES | YES | YES |
| R-Squared | 0.228 | 0.169 | 0.073 | 0.106 | 0.069 |
| Observations | 126 | 274 | 479 | 452 | 827 |

Note: The standard errors in parenthesis and clustered at the city and province level, and significance level are *** $p < 0.01$, ** $p < 0.05$, * $p < 0.10$.

At the firm level, larger firms with more robust internal control systems are able to grasp policy guidance more accurately [74]. Hence, to test the effect of firm size heterogeneity, this study divided the sample into large- and small-sized firms based on the median value of firm size in order to perform group regressions. The Table 9 presents the empirical results that credit expansion contributed significantly to the quality of green patents in large firms but not in small firms. This result verifies $H_3$ and might be because, during the post-crisis period, the primary focus of small-sized firms was on survival rather than upgrading their green innovation technologies. Furthermore, another possible reason is that large enterprises may be more resilient to market and technological risks and thus may be more inclined to undertake riskier green innovation projects in the face of city-level credit expansion. In contrast, small- and medium-sized enterprises are more risk-averse, especially during post-crisis periods. Overall, we find that urban credit expansion has a significant positive impact on green patent quality in large-sized firms compared to small-sized firms.

**Table 9.** The effect of credit expansion on the quality of green patents (firm size).

| Dependent Variable | Cit5Greenpatent | | Cit7Greenpatent | |
|---|---|---|---|---|
| | **Large firm** | **Small firm** | **Large firm** | **Small firm** |
| | **(1)** | **(2)** | **(3)** | **(4)** |
| *DID* | 1.228 *** | 0.128 | 1.289 *** | 0.086 |
| | (0.267) | (0.333) | (0.293) | (0.352) |
| *lnpgdp* | 0.331 | −0.381 | 0.159 | −0.356 |
| | (0.298) | (0.256) | (0.346) | (0.302) |
| *Fai* | −0.390 | −1.119 ** | −0.805 | −1.646 *** |
| | (0.488) | (0.481) | (0.503) | (0.577) |
| *Growth* | 0.039 | 0.010 | 0.061 * | 0.020 |
| | (0.026) | (0.033) | (0.031) | (0.039) |
| *Industry* | −2.917 ** | −0.209 | −1.808 | −0.051 |
| | (1.399) | (1.603) | (1.716) | (1.932) |
| *Investment* | −2.187 | 47.534 * | 1.155 | 44.936 |
| | (18.970) | (26.092) | (22.678) | (38.571) |
| *Volatility* | 0.533 | −0.393 | 0.538 | −0.598 * |
| | (0.325) | (0.287) | (0.366) | (0.319) |
| *Tangibility* | 6.881 ** | 5.919 ** | 8.602 ** | 6.804 ** |
| | (2.543) | (2.608) | (3.191) | (2.561) |
| *Leverage* | −0.339 | −0.429 | −0.184 | −0.649 |
| | (2.428) | (1.703) | (2.771) | (1.878) |

**Table 9.** *Cont.*

| Dependent Variable | Cit5Greenpatent | | Cit7Greenpatent | |
|---|---|---|---|---|
| | **Large firm** | **Small firm** | **Large firm** | **Small firm** |
| | **(1)** | **(2)** | **(3)** | **(4)** |
| *Profitability* | −1.822 | 1.987 | −0.135 | 1.878 |
| | (4.078) | (6.491) | (4.852) | (2.082) |
| *Size* | −0.210 | 1.345 ** | 0.477 | 1.442 ** |
| | (0.729) | (0.353) | (0.801) | (0.618) |
| Firm FE | YES | YES | YES | YES |
| Year FE | YES | YES | YES | YES |
| R-Squared | 0.490 | 0.292 | 0.476 | 0.286 |
| Observations | 547 | 718 | 547 | 718 |

Note: The standard errors in parenthesis and clustered at the city and province level, and significance level are *** $p < 0.01$, ** $p < 0.05$, * $p < 0.10$.

## 5. Robustness Test

We performed robustness tests in order to validate the results we obtained, as detailed in this section.

As the treatment and control groups were not randomly chosen, and the treatment and control groups presented different characteristics, self-selection bias may have occurred in the model. Furthermore, this bias can create correlations between the explanatory variables and the residual terms, giving rise to endogeneity. In order to address the self-selection bias and mitigate the potential endogeneity concern, we applied the PSM-DID estimation to conduct robustness tests. As a result, no significant difference in matching variables between the treatment group and the control group was observed after our treatment matching. Table 10 shows that the DID coefficients were still positively correlated with the quality of green patents at the 1% significance level. Overall, the PSM reduced the selection bias between the treatment and control groups and improved the accuracy of causal effect estimates. Accordingly, we obtained reliable estimation results through this method and suggest that urban credit expansion positively affects the quality of a firm's green patents.

**Table 10.** PSM-DID result.

| Dependent Variable | Cit5greenpatent | Cit5greenpatent |
|---|---|---|
| | **(1)** | **(2)** |
| *DID* | 0.509 *** | 0.538 *** |
| | (0.185) | (0.208) |
| *lnpgdp* | 0.033 | 0.013 |
| | (0.189) | (0.212) |
| *Fai* | −0.379 | −0.728 |
| | (0.397) | (0.453) |
| *Growth* | 0.024 | 0.047 ** |
| | (0.024) | (0.027) |
| *Industry* | −0.606 | −0.641 |
| | (1.239) | (1.390) |
| *Investment* | 12.462 | 22.390 |
| | (15.516) | (17.691) |
| *Volatility* | 0.148 | 0.163 |
| | (0.134) | (0.152) |
| *Tangibility* | 0.802 | 1.374 |
| | (1.556) | (1.774) |
| *Leverage* | 0.401 | 0.440 |
| | (1.076) | (1.227) |
| *Profitability* | 3.245 | 5.716 ** |
| | (2.377) | (2.710) |

**Table 10.** *Cont.*

| Dependent Variable | Cit5greenpatent | Cit5greenpatent |
|---|---|---|
| | **(1)** | **(2)** |
| *Size* | 0.040 | 0.044 |
| | (0.261) | (0.298) |
| Firm FE | YES | YES |
| Year FE | YES | YES |
| R-Squared | 0.295 | 0.329 |
| Observations | 1676 | 1676 |

Note: The standard errors in parenthesis and clustered at the city and province level, and significance level are *** $p < 0.01$, ** $p < 0.05$.

We next used the total average number of citations to green patents as the replacement dependent variable to re-test whether this effect was still valid (see Table 11). The robustness results show that the DID coefficients were 0.390 and 0.505, respectively, indicating a significant positive correlation with the quality of green patents at the 5% level. In other words, our conclusion that the implementation of urban credit expansion can significantly contribute to the quality of a firm's green innovations is robust.

**Table 11.** Replacement of the dependent variable.

| Depend Variable | Mean-Citations | Mean-Citations | PSM-DID |
|---|---|---|---|
| | **(1)** | **(2)** | **(3)** |
| *DID* | 0.390 ** | 0.505 ** | 0.540 * |
| | (0.187) | (0.223) | (0.324) |
| *lnpgdp* | | −0.025 | −0.453 |
| | | (0.200) | (0.278) |
| *Fai* | | −0.592 | −0.684 |
| | | (0.347) | (0.482) |
| *Growth* | | 0.017 | 0.052 * |
| | | (0.026) | (0.029) |
| *Industry* | | −0.140 | 1.391 |
| | | (1.115) | (1.867) |
| *Investment* | | 11.459 | 25.415 |
| | | (16.691) | (19.245) |
| *Volatility* | | −0.139 | 0.118 |
| | | (0.150) | (0.166) |
| *Tangibility* | | 2.955 * | 0.860 |
| | | (1.569) | (1.929) |
| *Leverage* | | −0.315 | 0.182 |
| | | (1.101) | (1.335) |
| *Profitability* | | 2.454 | 6.879 *** |
| | | (2.990) | (2.948) |
| *Size* | | 0.728 ** | 0.205 |
| | | (0.353) | (0.324) |
| Firm FE | YES | YES | YES |
| Year FE | YES | YES | YES |
| R-Squared | 0.330 | 0.321 | 0.285 |
| Observations | 1267 | 1267 | 1676 |

Note: The standard errors in parenthesis and clustered at the city and province level, and significance level are *** $p < 0.01$, ** $p < 0.05$, * $p < 0.10$.

## 6. Hypothesis and Dynamic Effects Test

It is necessary to check whether the parallel trend assumption is satisfied before applying the DID methodology. In the absence of a policy intervention, trends in the outcome variables should be consistent between the experimental and control groups, allowing for unbiased difference-in-difference estimates to be obtained. Thus, an analysis of the dynamic effects of green patent quality was conducted, following [75] and using the following model:

$$Y_{it} = \alpha + \sum_{-3 \leq j \leq 3} \beta_j Treat_{it} \times Time_{it} + \lambda X_{it} + \mu_i + r_t + \varepsilon_{it} \tag{4}$$

where $\beta_j$ indicates effects from the first three periods before policy intervention to the last three periods after policy implementation, and $\beta_0$ denotes the impact of the current treatment period. Therefore, the value of the DID term is 1 when the year is the current treatment period and 0 otherwise.

Table 12 presents the results of the dynamic effects analysis of green patent quality, which is a prerequisite for robust difference-in-difference estimations. Specifically, the coefficients for Pre-2 and Pre-3 were close to zero and insignificant for all of the green patent quality proxies, indicating that the trend of quality change between the low- and high-impact groups was similar before the impact of exogenous credit expansion. Meanwhile, the coefficients for Post-1 were positive and significant at the 5% level, indicating a significant difference in the trend of quality change between high- and low-impact groups during the economic stimulus period. Generally, this analysis demonstrates that the DID model in this work satisfied the parallel trend assumption.

**Table 12.** Dynamic effect test.

| Dependent Variable | Cit5greenpatents | Cit7greenpatents |
|---|---|---|
| | **(1)** | **(2)** |
| *Pre_3* | 0.126 | 0.293 |
| | (0.336) | (0.500) |
| *Pre_2* | 0.004 | −0.202 |
| | (0.383) | (0.547) |
| *Current* | −0.418 | −0.272 |
| | (0.532) | (0.663) |
| *Post_1* | 1.050 ** | 1.252 *** |
| | (0.511) | (0.634) |
| *Post_2* | 0.528 | −0.140 |
| | (0.503) | (1.115) |
| *Post_3* | 0.092 | 0.191 |
| | (0.370) | (0.519) |
| *2004 year* | 0.000 | 0.000 |
| | (0.000) | (0.000) |
| *2005 year* | 0.579 *** | −0.159 |
| | (0.103) | (0.119) |
| *2006 year* | 0.521 *** | 0.813 *** |
| | (0.107) | (0.116) |
| *2007 year* | 2.387 *** | 1.712 *** |
| | (0.162) | (0.180) |
| *2008 year* | 2.889 *** | 3.688 *** |
| | (0.231) | (0.354) |
| *2009 year* | 2.329 *** | 1.606 *** |
| | (0.319) | (0.363) |
| *2010 year* | 1.792 *** | 0.916 *** |
| | (0.238) | (0.261) |
| *2011 year* | 1.983 *** | 1.018 *** |
| | (0.258) | (0.280) |
| *2012 year* | 1.822 *** | 0.653 *** |
| | (0.217) | (0.228) |
| *2013 year* | 1.757 *** | 0.532 *** |
| | (0.188) | (0.195) |
| *2014 year* | 1.220 *** | −0.083 |
| | (0.181) | (0.188) |
| *2015 year* | 1.104 *** | −0.194 |
| | (0.175) | (0.183) |

**Table 12.** *Cont.*

| Dependent Variable | Cit5greenpatents | Cit7greenpatents |
|:---:|:---:|:---:|
| | **(1)** | **(2)** |
| *Constant* | 1.597 *** | 2.863 *** |
| | (0.179) | (0.261) |
| *Observations* | 2022 | 2046 |
| *R-squared* | 0.137 | 0.153 |

Note: Pre_3 and Pre_2 are dummy variables. If the observation is the data for year 3 and year 2 before the credit expansion shock, the indicator takes the value of 1, respectively, and 0 otherwise. This work drops Pre_1 to avoid the multicollinearity issue. If the observation is the data for the year of the credit expansion shock, the Current dummy variable takes the value of 1 and 0 otherwise. When the observation is the data for year 1, year 2 and year 3 after the policy shock, Post_1 and Post_2 take the value of 1, respectively, and 0 otherwise. Finally, *** and ** indicate the significance at 1% and 5% levels, respectively.

However, we find that the effect of this policy intervention seems to be short-lived, which may be due to the fact that most of the funds in the stimulus package went to large infrastructures. In order to accelerate project implementation, the government may have relaxed environmental protection standards or reduced the requirements for environmentally friendly technologies. This may have led to an impact on the improvement of the quality of green patents, as the development and implementation of environmentally friendly technologies require stringent environmental protection standards. In addition, stimulus packages are primarily focused on rapidly expanding domestic demand and boosting economic growth, so governments are more inclined to support projects that directly create jobs and increase output. Areas that are more likely to yield economic benefits in the short term will receive more attention than green technologies and innovations.

## 7. Conclusions and Discussion

Despite the controversy surrounding the credit expansion, the CNY 4 trillion investment plan has been effective in stimulating economic growth [76]. Nevertheless, the credit expansion has indirectly led to accelerated growth in the production of major energy-consuming manufacturing industries since 2009, which will undoubtedly put pressure on the fulfilment of energy-saving and carbon emissions reduction targets. Calculations show that the four trillion CNY investment will increase energy consumption by a total of about 113 million tonnes of standard coal, equivalent to 260 million tonnes of carbon emissions, with an average annual increase of 130 million tonnes. Furthermore, scientific and technological innovation is an important means of guaranteeing energy conservation and carbon emissions reduction [77], requiring the government to promote research and development at the national level. For example, former U.S. President Barack Obama announced in August 2009 that he would take out USD 2.4 billion from the USD 787 billion economic stimulus package to support the research and development of new energy vehicles. Although CNY 370 billion of the CNY 4 trillion investment programme was allocated to structural adjustment and innovation, which will have a positive effect on the technological upgrading of firms and thus on the reduction in energy consumption per unit of output, China can be seen as placing "emphasis on construction while light on R&D" in the field of energy saving and carbon emission reduction, especially considering that its independent intellectual property rights in research and development are obviously insufficient, when compared with those of other developed countries.

We provide evidence that supports the previous literature that credit expansion can promote corporate innovation. We further illustrate how credit expansion in 2009 affects the quality of green patents from the perspective of urban credit expansion. Our empirical tests validate the hypotheses, and we also test the impact at different tiers of cities and for different sizes of firms. This research took the economic stimulus package in China as a quasi-natural experiment to analyse the impact of credit expansion on the quality of green innovation. In general, we show similar findings to that presented in [43,78]. Our empirical results suggest that urban credit expansion has a significant positive impact on firms'

ability to strengthen their green patent quality, and this effect is particularly significant in first-tier cities and for large-sized firms. Moreover, *this* research confirmed that the credit expansion programme contributed to enhancing the quality of green innovation, with cities that received a larger share of loans from the credit expansion creating higher quality green patents than cities receiving a smaller share of loans from the credit expansion. Thus, while the environmental governance capacity of first-tier cities should be brought into full play, the sensitivity of lower-tier cities to credit expansion policies should also be considered and strengthened through actions such as unblocking the policy transmission of the financial system and incentivising the engagement of companies. Through the direct expansion of financing channels, the credit expansion policy can reduce the cost of financing, such that funds injected directly into the field can help improve the quality of green patents issued by enterprises, promoting economic development while realising green transformation. By studying the impact of the credit expansion policy on the quality of corporate green patents, we contribute reference material and inspiration for the credit expansion policy and green finance in China and, to a certain extent, fill the gaps in the existing literature.

Notably, 81% of the credit expansion funded in the stimulus package was invested into the construction of large-scale infrastructure with coal, cement, iron, and steel as the main consumables, thus putting huge pressure on carbon emissions. According to the Global Green and Low Carbon Technology Patent Statistical Analysis Report (2023), the average annual growth rates of patents granted by Chinese patentees in the fields of clean and efficient use of coal and cleaner use of oil and natural gas were 9.0% and 4.6%, higher than the global averages of 8.5 and 6.5 percentage points, respectively. Therefore, against the background of persistently high carbon emissions from fossil energy, China has added innovative momentum to the green development of global fossil energy. This report also supports our finding that credit expansion significantly improves the quality of green patents, which can be effectively confirmed in the fossil energy industry. In addition, this report pointed out that, due to the obvious presence of large centralised enterprises and research institutes, China's green and low-carbon patent grants were led by the cities of Beijing, Guangdong, and Jiangsu; in particular, the top three provinces and cities in China, in terms of green and low-carbon patent grants from 2016 to 2022, were Beijing (23,300), Guangdong (21,200), and Jiangsu (19,500), which together accounted for 40.7% of the total number of patents granted in China. Our results indicate that credit expansion is more significant in improving green patent quality in first-tier cities and large enterprises, confirming the hypotheses of this report and the related positive effect in the real world. Overall, China has become an important driving force for global green and low-carbon technological innovation in the face of a general slowdown in global green and low-carbon technological innovation.

In conclusion, green finance takes on the role of not only creating a new driving force for economic expansion but also speeding up the green transformation by facilitating green development and economic structural adjustment [79]. This paper argues that the government should improve the targeting of credit expansion policies to channel the effects obtained through the green economy. The government should also formulate flexible and targeted policies for enterprises in different ownership systems and regions. Simultaneously, it is necessary to consider the promotion of financial marketisation more actively, raising efficiency in the allocation of financial resources, improving relevant supporting policies, and providing financial protection for the innovative activities of enterprises [80]. Specifically, financial resources should be weighted more towards the field of green innovation, which also means improving the efficiency of the use of funds, thereby achieving both efficient financial support for green innovation and sustainable development of the financial industry [81].

Finally, our analysis provided important insights for implementing credit expansion policies and achieving sustainable development in China and other emerging countries. We suggest that investments in environmental protection and other areas must be combined with investments in economic construction to achieve a harmonious development of the

economy and the environment. While this paper explored the effect of implementing urban-level credit expansion on the quality of green patents in China, there were still some shortcomings. First, the analysis of the paths and mechanisms through which credit expansion affects the quality of green patents was relatively shallow and could be enhanced through the inclusion of additional information, such as the channels through financing constraints and signal-releasing effects. Second, the PSM model is mainly applicable to cross-sectional data, while the DID model is mainly applicable to panel data, and the difference in the scope of application of the two models is a fundamental issue affecting the validity of the PSM-DID model. Future research will aim to optimise the empirical methods. Third, this study focused on the patent quality of Chinese listed companies but did not examine the heterogeneity of listed companies (e.g., differences in geographical distribution and industry). Fourth, as this study focuses on China, the findings and implications may not be generalisable to other countries with different economic and political structures. Future research will take into account the characteristics of different countries in their analyses. These contexts will be improved in future research.

**Author Contributions:** Conceptualization, Z.S. and J.T.; methodology, Z.S.; software, Z.S.; validation, J.T., Z.S. and H.Z.; formal analysis, Z.S.; investigation, Z.S.; resources, Z.S.; data curation, F.P.; writing—original draft preparation, Z.S.; writing—review and editing, J.T.; visualization, H.Z.; supervision, H.Z.; project administration, H.Z.; funding acquisition, H.Z. All authors have read and agreed to the published version of the manuscript.

**Funding:** This study is not externally funded.

**Institutional Review Board Statement:** The study is not ethically relevant and do not require ethical approval.

**Informed Consent Statement:** The study do not involve humans.

**Data Availability Statement:** The data provided in this study as well as the Stata code are available upon request from the corresponding author.

**Conflicts of Interest:** The authors declare no potential conflicts of interest with respect to the research, authorship, and publication of this article.

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
