# Peer review of "How Urban-Level Credit Expansion Affects the Quality of Green Innovation: Evidence from China"

_sustainability, doi:10.3390/su16051725_

Round 1
Reviewer 1 Report
Comments and Suggestions for Authors
See attachment.

The paper is well organized and written, due to the current contents of the paper, the English writing should be carefully revised for clarity and concision. In addition, the authors also need to check potential grammar, expression and typos in the main text.
Reviewer 2 Report
Comments and Suggestions for Authors
Good morning, Dear Authors
The article is interesting, but I have a few comments that would improve its quality:
1. The data taken for analysis are too old (2004-2015, in some cases reference only to 2009). "Finally, this paper argues that China's credit expansion policy in 2009 has played a 19 role in improving the quality of green innovation and improve green transformation." If the policy dates back to 2009, how are the years 2004-2008 examined? Perhaps it would be worth comparing these two periods?
2. In the article, in line #47-50 it is indicated "It is expected that in the next few years will drive the social investment of 407.9 billion yuan, which is gradually put into the market of funds will improve the energy-saving industry to promote the rapid development of long-term." The note concerns the reference to time. The sentence has the future tense "expected", but it would be worth using "was expected". Which also suggests the possibility of checking whether the forecasts actually came true. In some parts of the article it looks like the article was written before the 2015 doc.
3. The authors suggest that "At the same time, it is necessary to more actively consider promoting financial marketisation, optimizing the allocation of financial resources, improving relevant supporting policies, and providing financial protection for the innovative activities of enterprises", but they do not say about what such a structure should look like after optimization. What would result from the research conducted?
4. It is doubtful whether the connection between credit expansion and the quality of green investments is confirmed. "Moreover, this research confirms that the credit expansion program contributes well to the quality of green innovation, with cities that received a larger share of loans from the credit expansion creating higher quality green patents than cities that received a smaller share of loans from the credit expansion". Rather, the article proves that economic growth has been maintained thanks to such expansion. Even if it is the number of certificates or credits, it does not talk about quality.
5. Not all data included in the analysis are statistically significant, perhaps it would be worth reducing the mathematical formula in lines 346-348 with unnecessary indicators. And present the updated formula at the very end of the results.
Reviewer 3 Report
Comments and Suggestions for Authors
Thank you for the opportunity to read this interesting paper. In the following, some aspects that could be improved are detailed. My opinions are constructive and not critical of the authors' work.
The structure of the paper is missing from the Introduction.
I assume that the authors insist on the conclusions of a 14-year-old report "The Impact of 37 China's Economic Stimulus Plan on Climate and Energy", from 2011, to give a context to the research, which refers to the period 2004-2015. However, why do they use the future tense in these paragraphs? The relevance of the predictions from that time (lines 51-55) would be greater if evidence showed if they came true. It would be more interesting to see what the results were, for example the ones expected for 2020, compared to the predicted ones.
Hypothesis H1: “Urban credit expansion significantly promoting the quality of green patents”, should be rephrased as H1: Urban credit expansion significantly promotes the quality of green patents. The entire paper must be carefully proofread because some phrases are quite difficult to understand.
There is no review of the literature regarding the citation of patents as a proxy for the quality of green innovation. The development of hypotheses is mainly based on the literature regarding the impact of credit expansion on innovation in general. Citation of patents appears directly in the hypotheses, without prior justification. Nor is the literature in the field of urban credit expansion sufficiently analyzed.
No details are provided about the 5 tiers in table 2, which cities are included, what they have in common from an economic point of view, environmental development, etc. It would help to interpret the results and formulate conclusions.
In section 3.2, only 2 sources are cited for the variables used in the models. A better connection with the existing literature is necessary in this part of the paper to capture its originality.
Descriptive statistics are superficially commented on. Reference is made only to the number of citations of green patents, the rest of the variables, including credit expansion, are not commented on at all.
The lack of a good connection with the literature is also observed in the interpretation of the results, where there is almost no reference to similar or different results from other studies. This creates a break between the beginning part (reviewing the literature and developing the hypotheses) and the ending part (results) of the paper.
Section 5 starts too suddenly with table 10, without any introductory phrase. Section 6, which describes a stage prior to the application of DID, should appear earlier in the paper, shouldn’t it? Before the application of the DID maybe...
The figures in the paper regarding the level of emissions, investments, etc. are not presented temporally (to what period do they refer to?) nor is their source given, so that they can be verified.
In the conclusions, no limits of the research are presented.
Good luck with the revision!
Comments on the Quality of English LanguageThe entire paper must be carefully proofread because some phrases are quite difficult to understand.
Reviewer 4 Report
Comments and Suggestions for Authors
Dear authors,
The topic of the manuscript is interesting and addresses the effect of urban credit expansion on the quality of green innovation in China.
In my opinion, the authors must enhance the quality of the article before publishing.
First, the authors must explain what they mean by the quality of green innovation and how the average number of citations to green patents (which reflects quantity and capacity to generate innovation) is appropriate for describing the quality of green innovation.
Also, if they used as a dependent variable the quality of green patents (expressed by the average number of citations to green patents), why did they use also the volatility of a firm's green innovation output (volatilitygreen) (that is expressed as the number of green patents filed by the firm in year t as well as the average number of applications filed by the firm over the next five years) as control variable? These variables seem similar.
The authors did not present the research method, why was used a fixed effect model? How can be explained the low R-square results?
The reference list does not comply with the journal guidelines and must be revised.
Comments on the Quality of English LanguageThe manuscript contains minor grammatical errors and some typos that should be checked carefully and corrected.
Reviewer 5 Report
Comments and Suggestions for Authors
Please see attached file.

Round 2
Reviewer 1 Report
Comments and Suggestions for Authors
The authors have revised the paper following previous concerns, and now I suggest an acceptance at this version.
Comments on the Quality of English LanguageThe English is fine at this version.
Author Response
We are glad to hear that you are satisfied with this version.
Thank you very much for your suggestions to improve the quality of this article!
We wish you all the best!
Reviewer 2 Report
Comments and Suggestions for Authors
Good morning, Dear Authors
I see that you have improved the article, but I think that the number of things to improve is only increasing. Please work on it seriously from a substantive point of view. Some issues conflict with each other. For example:
Text is missing before reference number 5.
In response to comment number 1, you suggest that the government's policy can be considered a shock to the economy. As a rule, policy must serve to support and strengthen the economy. Please think about this matter.
In your response you indicate that "Moreover, the reason we use old data is that we consider that the sample period for the DID test should not be too long". I do not agree with this justification.
Also, there was no satisfactory response to the remaining comments sent during the first review.
Kind regards.
Author Response
Thank you very much for your suggestions to improve the quality of this article, we take your comments very carefully and seriously.
Please see the attachment, thanks a lot!

Reviewer 3 Report
Comments and Suggestions for Authors
The authors mostly followed the previous recommendations. As the result, the paper is significantly improved.
Congratulations!
Comments on the Quality of English LanguagePaper still needs careful proofreading.
Author Response
We are glad to hear that you are satisfied with this version.
We've proofread the entire revised manuscript and will do proofreading again if necessary using the services recommended by the publisher.
Thank you very much for your suggestions to improve the quality of this article!
We wish you all the best!
Reviewer 5 Report
Comments and Suggestions for Authors
Thank you for responding to all of my comments. I am now satisfied with this version. I recommend the publication of the paper.
Author Response

(The authors gave the same response as above.)

Round 3
Reviewer 2 Report
Comments and Suggestions for Authors
Dear Authors
Unfortunately, I cannot recommend the article for publication in this magazine. In addition to the interesting research periods, the article also has substantive errors. For example:
"While the coefficients on Post-1 is positive and significant at 5% level, this implies that a significant difference in the trend of quality change between high and low impact groups during the economic stimulus period"
But in Figure 1, Post-1 ranges from 0 to 2. There are also elements indicating the negative impact of state policy in this area. Generally, you "praise" this approach in the article, but the final results do not confirm the text. Also, it is not known what Figure 1 shows.
In the final results, the authors also conclude that the goal has not been achieved and the whole thing requires a deeper analysis. There is no possibility to use the test results in other cases.
For me, the following conclusion also has no substantive value in the context of a given study:
"Third, this study focuses on the patent quality of Chinese listed companies but does not examine the heterogeneity of listed companies (different geographical distribution and industries)"
Kind regards
Author Response
Dear Reviewer,
We apologize that our research and subsequent responses were not to your satisfaction, we have endeavored to complete the study and were delighted to discuss a range of issues to improve the quality of this article. We will carefully consider your suggestions for research in the future.
Please allow us to thank you one last time for your helpful siggestions and wish you all the best for the future!
Best wishes.
The Authors